# MicrobeTrace: Retooling molecular epidemiology for rapid public health response

Ellsworth M. Campbell[1]☯*, Anthony Boyles[2]☯, Anupama Shankar[1], Jay Kim[2], Sergey Knyazev[1,3,4], Roxana Cintron[1], William M. Switzer[1]

**1** Centers for Disease Control and Prevention, Atlanta, Georgia, United States of America, **2** Northrup Grumman, Atlanta, Georgia, United States of America, **3** Oak Ridge Institute for Science and Education, Oak Ridge, Tennessee, United States of America, **4** Department of Computer Science, Georgia State University, Atlanta, Georgia, United States of America

☯ These authors contributed equally to this work.
* ells.cubed@gmail.com

**Data Availability Statement:** All relevant data are within the manuscript and its Supporting Information files.

**Funding:** The author(s) received no specific funding for this work.

## Abstract

Outbreak investigations use data from interviews, healthcare providers, laboratories and surveillance systems. However, integrated use of data from multiple sources requires a patchwork of software that present challenges in usability, interoperability, confidentiality, and cost. Rapid integration, visualization and analysis of data from multiple sources can guide effective public health interventions. We developed MicrobeTrace to facilitate rapid public health responses by overcoming barriers to data integration and exploration in molecular epidemiology. MicrobeTrace is a web-based, client-side, JavaScript application (https://microbetrace.cdc.gov) that runs in Chromium-based browsers and remains fully operational without an internet connection. Using publicly available data, we demonstrate the analysis of viral genetic distance networks and introduce a novel approach to minimum spanning trees that simplifies results. We also illustrate the potential utility of MicrobeTrace in support of contact tracing by analyzing and displaying data from an outbreak of SARS-CoV-2 in South Korea in early 2020. MicrobeTrace is developed and actively maintained by the Centers for Disease Control and Prevention. Users can email microbetrace@cdc.gov for support. The source code is available at https://github.com/cdcgov/microbetrace.

## Author summary

Rapid advances in the fields of data science and bioinformatics have significantly improved molecular epidemiology tools used in public health and have led to major changes in the way outbreak investigation and pathogen transmission studies are conducted. However, the need for specialized computer skills often impedes the use of many of these tools in the public heath domain. We bridge this knowledge gap by development of an intuitive, standalone tool called MicrobeTrace to securely integrate, visualize and explore pathogen epidemiologic data. MicrobeTrace is an easy to use browser-based tool which can effectively merge contact tracing and/or microbial genomic data with demographic or behavioral information, resulting in elegant and informative networks as well

**Competing interests:** The authors have declared that no competing interests exist.

as multiple customizable visualizations. MicrobeTrace can be used offline, with analyses being performed locally in the field, ensuring secure and confidential use of personally identifiable information (PII). We provide real world examples of how MicrobeTrace has been used in public health, including COVID outbreak investigations.

## Introduction

The burgeoning field of public health bioinformatics has given rise to a plethora of specialized software for analysis and visualization of pathogen genomic data to aid outbreak investigations [1,2]. Implementation of these analytic tools can be complex and fraught with technical and administrative barriers [3]. Historically, many public health workers with educational backgrounds in medicine, epidemiology, and laboratory sciences lack informatics skills needed to collect, analyze and display data ([4]; *Applications of Clinical Microbial Next-Generation Sequencing: Report on an American Academy of Microbiology Colloquium held in Washington, DC, in April 2015*), hindering the routine use of bioinformatic tools in public health, especially for molecular epidemiology investigations. This skill mismatch tends to be more pronounced at local health departments, with limited capacity and funding for informatics infrastructure [5].

Technical and administrative barriers are often reduced by moving complex analytics and computation to off-site servers. However, state public health laws often prohibit cloud use for the storage of sensitive data. Tool accessibility can also be hampered by cluttered user interfaces [6–9] and unwieldy workflows [4,10–12]. Given the breadth of genetic sequencing technologies and bioinformatic methods, bioinformatic tools should ideally be secure, easy to use, and capable of accepting or exporting data in commonly used formats to facilitate rapid public health investigation and response.

To this end, we developed a browser-based, standalone tool to integrate, visualize and explore data collected during outbreaks and investigations of transmission clusters such as demographic and behavioral information, high-risk contact lists and microbial genomic data. MicrobeTrace was designed to construct pathogen genetic distance networks and visually integrate them with contact tracing networks to better characterize transmission. In contrast with other commonly used tools [10,12], visual attributes like size, shape and color can be modified by the user via simple interactions in real-time, without modification of the underlying data. MicrobeTrace is well suited to working with sensitive personally identifiable information (PII) because it performs all computations locally and does not transmit any data. When using a supported and updated web browser (e.g., Chrome, Firefox, or Edge), cached files are cleared when the browser session ends. MicrobeTrace is available at the Centers for Disease Control and Prevention (CDC) Github website and thereafter used on the user's computer without an internet connection, making it ideal for field use.

Here, we describe the utility of MicrobeTrace across multiple public health use cases; outbreak response and transmission analysis for a broad spectrum of infectious diseases, such as tuberculosis, viral hepatitis, sexually transmitted diseases and special pathogens like SARS-CoV-2 and Ebola.

## Design and implementation

MicrobeTrace has been developed according to an agile open source model and guided by input from public health practitioners. End users of HIV-TRACE were integral in the development of the minimum viable product and initial release of MicrobeTrace in June 2018. Since

then, end users at federal, state, and local levels have been critically important in the development, testing, and addition of all features. Engagement with end users occurs on a continuous basis through multiple modalities, from email and instant messaging to screenshare sessions, in-person seminars, and webinars. The most valuable feedback in refining MicrobeTrace has been derived from screenshare sessions with end users at state and local levels who are engaging with the tool during an active outbreak investigation.

All code is available via https://github.com/CDCgov/MicrobeTrace [13], enabling users to submit and monitor feature requests and system bug reports. All code is indexed by the federal open source repository [14] and promoted by Code.gov [15]. The MicrobeTrace codebase is regularly scanned by Fortify Software [16] and SonarQube [17] to ensure security and code stability. Modules of code that depend on each other are automatically monitored for vulnerabilities and updated by GitHub's Dependabot service, ensuring that security vulnerabilities are rapidly detected, reported to our development team, and addressed. GitHub's *Actions* service is used to automate the process of testing newly developed features before official release to ensure that each time new features are added into MicrobeTrace, all pre-existing functionality is automatically tested prior to an official release. In addition to a detailed user manual, training is provided through three modalities: (1) small *ad-hoc* webinar sessions (5–20 attendees) to support specific outbreak and cluster investigations, (2) large in-person training sessions (20–100+), and (3) a recorded webinar available via YouTube [18]. A detailed user manual and example data files are available at https://github.com/CDCgov/MicrobeTrace/wiki.

## Results

MicrobeTrace handles a variety of file types and formats traditionally collected during public health investigations (Fig 1). Pathogen genomic information can be integrated as raw genomic sequences, genetic distance matrices, pairwise genetic distances, or phylogenetic trees (Newick files). Epidemiologic and other metadata about cases (node lists) and their high-risk contacts (edge or link lists) can be integrated as spreadsheets. Importable in a variety of file formats, these file types can be visualized independently or in-concert to achieve different analytic goals. The ability to integrate genomic and epidemiological data gives the user a more holistic picture of an ongoing public health investigation.

MicrobeTrace is well adapted for public health because of confidential but effective use of sensitive data collected during outbreak investigations. Most web-based bioinformatic applications require submission of data over the internet for processing by a remote *server-side* application before results can be returned to the user. In contrast. MicrobeTrace is a *client-side only* application which achieves local processing through open source development and modifications of existing algorithms to align [19–21], compare [4,22,23], and evaluate genomic sequences and their relationships to one another [24–27], giving results that are interchangeable with those derived from their native implementations. A novel extension of the network evaluation method is described below as the 'Nearest Connected Neighbor'.

PII and other sensitive information like geospatial coordinates, zip codes, and phone numbers should only be accessible to Disease Investigation Specialists conducting contact tracing interviews. However, an epidemiologist performing a retrospective analysis can use the same visualization layout with remapped labels, colors, shapes and sizes. Indeed, sensitive geocoordinates can still be used to produce informative maps by applying the random 'jitter' function in MicrobeTrace to reduce the precision of the displayed map marker. In concert, these diverse and accessible controls enable public health experts to securely leverage sensitive data without confidentiality risks.

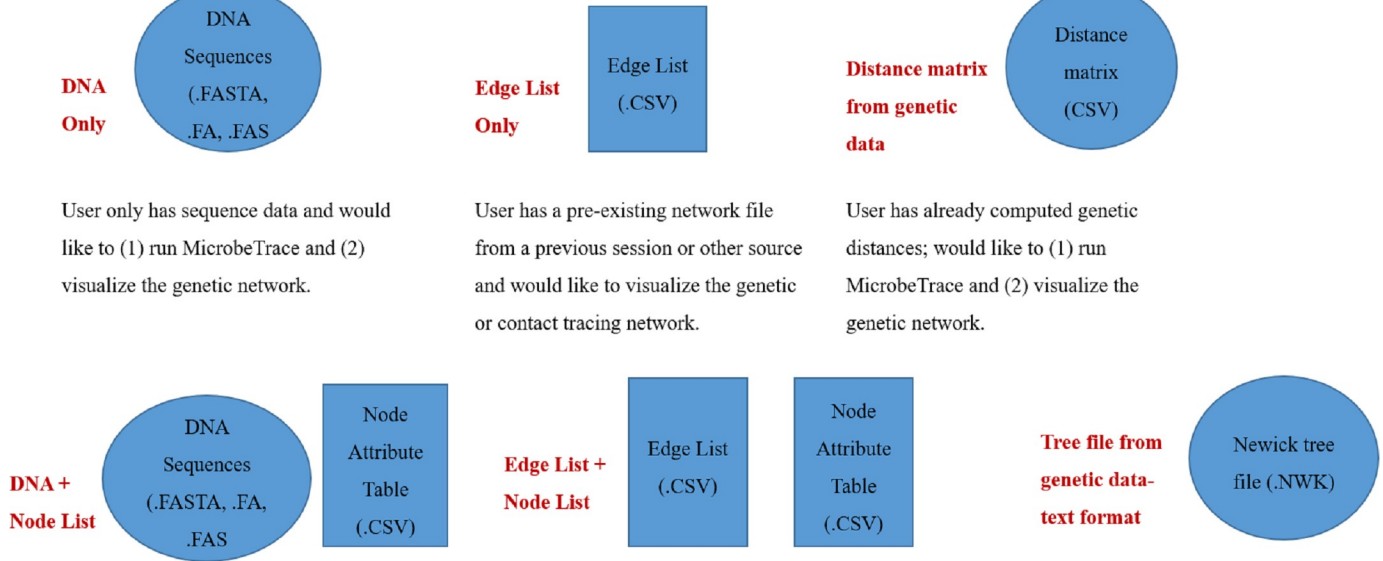

**Fig 1. MicrobeTrace accepts input data in a variety of formats.** This figure displays the most common use cases and their required files.

We demonstrate the bioinformatics capacity of MicrobeTrace using a publicly available HIV-1 dataset consisting of 1,164 sequences of the partial polymerase (*pol*) region from a recent study in Germany with associated metadata describing behavioral risk factors and gender [28]. The bioinformatics workflow to build genetic distance networks in MicrobeTrace begins with a pairwise sequence alignment of each input sequence against a reference of choice, according to the Smith-Waterman algorithm [19–21]. Multiple sequence alignments are time consuming and are not used. A user can align to a curated reference, an arbitrary custom reference, or the first input sequence. Once aligned, pairwise genetic distances are calculated according to either a raw Hamming distance (for SNPs) or the Tamura-Nei substitution model (TN93) for sequences [4, 22, 23]. With the TN93 substitution model users can select configuration of ambiguous bases as previously described [4]. Notably, users are provided the option to select the distance threshold value that best fits their pathogen or public health use case [29], in this case for HIV-1 1.5% nucleotide substitutions per site (Fig 2A). The initial distance threshold can easily be changed after the first genetic network construction to explore the effect of different thresholds on the network. MicrobeTrace also offers the ability to filter by cluster size thresholds in the same 'Global Settings' menu. Here, we have filtered for clusters of size N ≥ 5 after the 1.5% genetic distance threshold is applied. This filter hides 73.1% (N = 851) sequences that are too genetically distant to cluster with any other sequences in the dataset as well as 17.9% (N = 208) of individuals whose HIV-1 sequences reside in clusters of size N ≤ 4. HIV-1 sequences from the remaining 9.0% (N = 105) of individuals are displayed as genetic distance networks (Fig 2). Variables of interest can be readily mapped to nodes or links, including HIV-1 *pol* drug resistance mutations to identify clusters of transmitted drug resistance (Fig 2C).

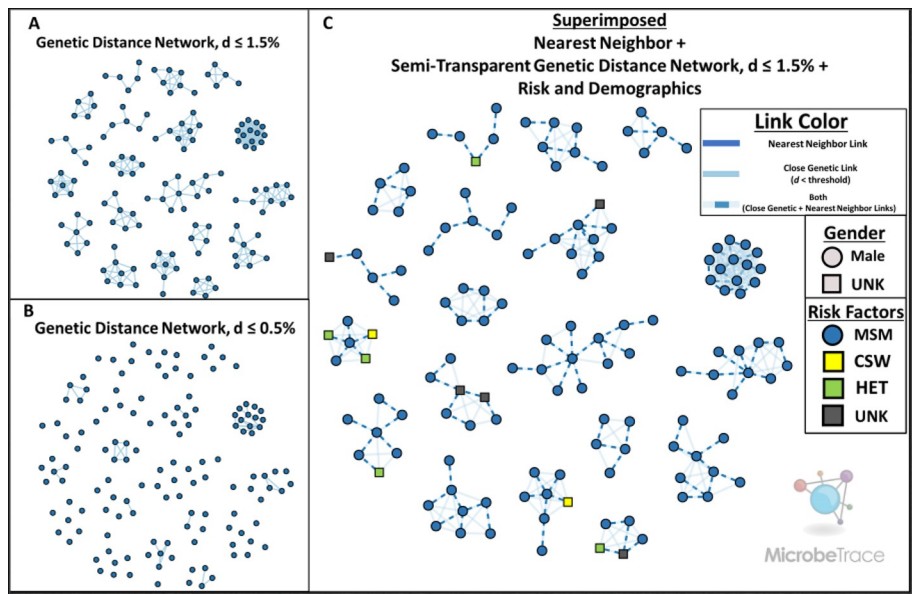

**Fig 2. MicrobeTrace excels at rendering pathogen genetic distance networks and mapping visual characteristics to user-provided metadata.** (**2A**) The HIV-1 partial polymerase (pol) distance network, with a genetic distance threshold (d) of 1.5%. (**2B**) The same HIV-1 pol network shown in 3A with node positions held constant, but with a more stringent genetic distance threshold (d) of 0.5%. (**2C**) The same HIV-1 pol network shown in 3A with node positions held constant. Nearest connected neighbor links have been superimposed as dashed lines. The transparency of links that do not connect nearest neighbors has been increased. Gender and transmission risk factors have been mapped to node shape and color, respectively.

## Scalability

The scalability of MicrobeTrace is limited by the user's computer configuration (RAM, processor, etc.) and the web browser used. We tested the scalability of MicrobeTrace to render a network in Google Chrome version 91.4 on a 3.7 GHz core_i7 processor with a 1,024 GB hard drive and 16 GB RAM, using fasta files containing aligned partial HIV-1 *pol* sequences (1,300-bp) and complete SARS-CoV-2 genomes (30,000-bp) of various totals. We also determined the speed of network generation by using Newick files with different numbers of taxa. The totals used in these analyses are larger than those typically seen in local outbreak investigations. S2, S3 and S4 Tables and S2 Fig show the computation times for these analyses. MicrobeTrace could easily render a network for 5,000 HIV-1 *pol* sequences in about 2.7 minutes compared to 1.5 minutes for 1,000 complete SARS-CoV-2 genomes. For 1,500 SARS-CoV-2 complete genomes MicrobeTrace froze and could not render a network. It took about 7.2 minutes to render the network for a Newick tree file containing 1,250 taxa. For larger genomes, like SARS-CoV-2 and those from bacterial pathogens, alignments of single nucleotide polymorphisms (SNPs) could be used to decrease the network rendering times [30].

## Pre-computed genetic distance networks

A simple nucleotide substitution model is not always suitable to understand phylogenetic relationships. Rather than require the use of a single model, MicrobeTrace supports the integration of precomputed distance matrices and pairwise distance lists. For distance matrices, both full matrix and PHYLIP formats are accepted. MicrobeTrace also provides a novel and simple filtering algorithm to render only the nearest connected genetic neighbor(s) for each node, while still maintaining cluster connectivity. Where two genetically equidistant neighbors are

possible, both links are rendered when the 'Nearest Connected Neighbor' filter is applied. This approach is particularly useful to understand the historical context of an entire cluster, while focusing on the part of the cluster exhibiting the most concerning and rapid growth. For example, an HIV-1 cluster in rural southeastern Indiana grew rapidly in 2015 but underwent slow growth for nearly a decade prior [31]. The nearest connect neighbor method yields results similar to a non-exhaustive search for all minimum spanning trees, as has been previously described [31,32]. The threshold and nearest connected neighbor filters are not mutually exclusive and can therefore be applied simultaneously to ensure that genetically distant nodes remain disconnected. This enables the inclusion of related, but more distant sequences in a cluster visualization while minimizing the information overload typically seen with less stringent distance thresholds (Fig 2A). Genetic distance links that fell below the 1.5% threshold but were not included as a nearest connected neighbor link are shown at reduced opacity (Fig 2C).

## Patristic distance networks

Phylogenies are ubiquitous in public health and bioinformatics but may be difficult to integrate with more traditional contact tracing data. While powerful new tools are available to integrate taxon-level characteristics into phylogenies, integration of paired contacts is unavailable. MicrobeTrace overcomes this hurdle by traversing the phylogeny as a standard Newick file to calculate and render the corresponding pairwise patristic distance network. Specifically, these are tip-to-tip measurements between individuals on an evolutionary tree that account for the most recent common ancestor. Users can generate the Newick files using their favorite phylogenetic methods and pathogen specific nucleotide substitution models.

## Epidemiologic networks

Importantly, MicrobeTrace supports the visualization of not just sequence data, but also data routinely collected during contact tracing during an outbreak or cluster investigation. Acceptable data are not limited to person-to-person links and can include person-to-place or place-to-place links. To visually differentiate persons from places, MicrobeTrace can style the shape of any network node according to a node type column (e.g., node Type = 'Person' or 'Place') defined in the data set. If additional metadata are available to describe a link, it can be colored according to user-defined categorical variables. Alternatively, an option is provided to scale link width according to a user-defined numeric variable or its reciprocal.

## Multi-layer networks

Epidemiologic and genetic networks often offer complementary perspectives about transmission clusters [33]. MicrobeTrace can render an arbitrary number of networks simultaneously by representing multiple overlapping links between pairs of nodes (e.g., hyperlinks) as color-mapped, dashed lines. In addition to independent color-mappings according to underlying data, the effect of a particular network layer can either be hidden or accentuated via independent transparency controls. For example, to protect individual-level privacy, public health experts may choose to make epidemiologic reports of high-risk contact invisible while rendering only close genetic links when producing figures for public consumption.

## Maps with network overlay

Integrated epidemiologic and genetic networks can be used to inform policy and prevention efforts when augmented with additional information. MicrobeTrace can generate choropleth maps, globe diagrams, or more common map projections. MicrobeTrace can display high-

resolution geospatial map tiles from a JavaScript map service Stamen.com [34]. If internet access is unavailable, MicrobeTrace mapping functions offline with pre-computed shapefiles describing countries, as well as U.S. states and counties. MicrobeTrace also enables users to contextualize their maps with a network overlay that maintains all color mappings defined in the network visualization. Users can select from various geographic units, ranging from country, down to zip code for the U.S. or paired latitude and longitude values. For each geographic level, a marker is placed at the geographic centroid. Over-plotting can be addressed by a combination of automated aggregation or manual transparency tools. Maps can also be customized with user-provided geospatial data in the GeoJSON format.

## Customization and interactive exploration

To demonstrate the visualization capacity of MicrobeTrace, we present a publicly available data set describing clinical, demographic and contact tracing data derived from the investigation of the Korean COVID-19 outbreak [35]. The data set does not contain coronavirus sequence data, but instead details 383 transmission histories between 510 cases. It also contains an additional 1,627 cases of COVID-19 with no documented transmission histories. As before, using filtering capabilities unique to MicrobeTrace, we limit our visualizations to transmission clusters of size ≥ 5 cases (Fig 3).

MicrobeTrace is centered around integration and visualization of pathogen genomic and network data but offers an array of customizable tables, charts, and geospatial maps that

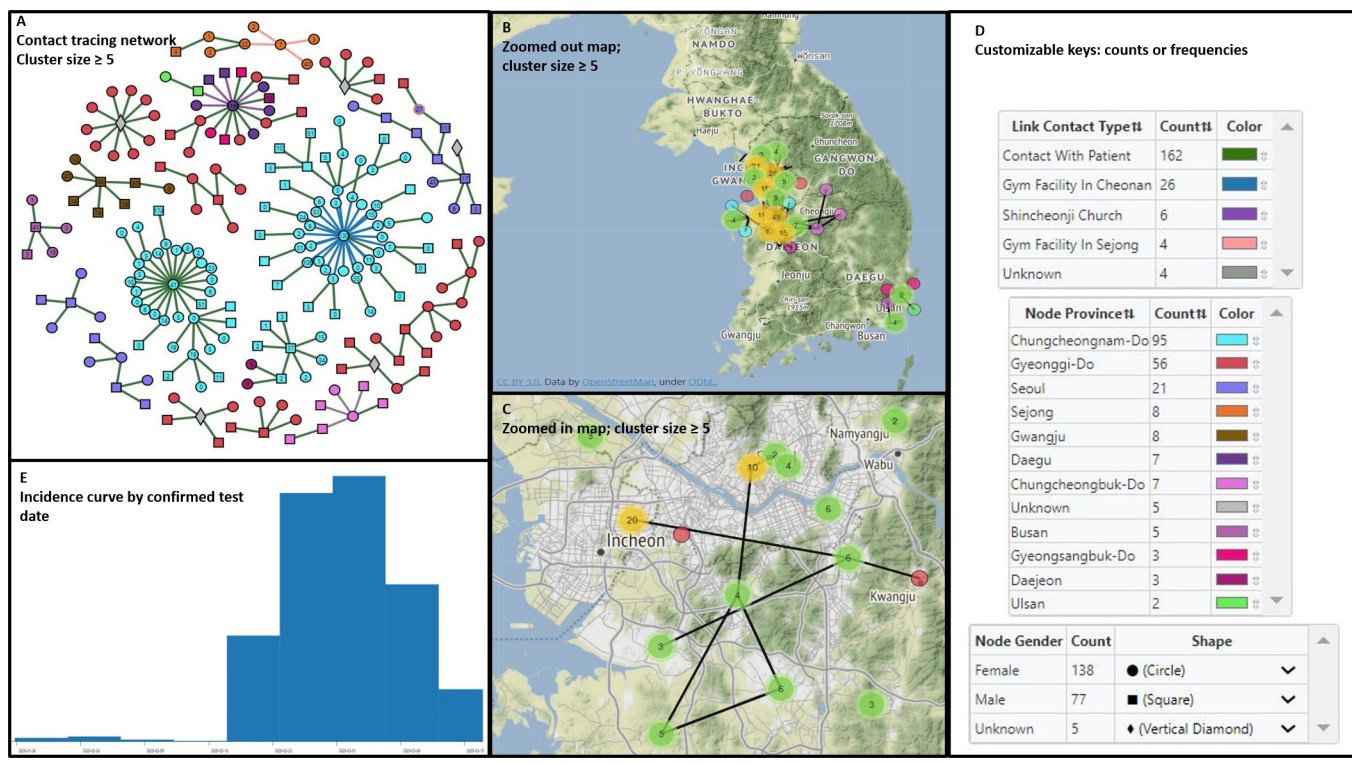

**Fig 3. MicrobeTrace allows the creation of informative dashboard visualizations by tiling of different views within the browser window. (3A)** Reports of high-risk contact between COVID-19 cases in clusters of size N ≥ 5, nodes are (i) colored by province, (ii) shaped by gender, and (iii) labeled with the total number of high-risk contacts. **(3B)** Geospatial map of clusters of size N ≥ 5 zoomed to show only Seoul, South Korea. **(3C)** Geospatial map of clusters of size N ≥ 2. Node positions have been randomly altered with the 'jitter' functionality to preserve patient privacy. **(3D)** In-application color and shape keys that offer interactive color-pickers and labeling. **(3E)** Incidence curve showing confirmed test date. Map tiles provided by http://stamen.com/ under CC BY 3.0; data by openstreetmaps.org

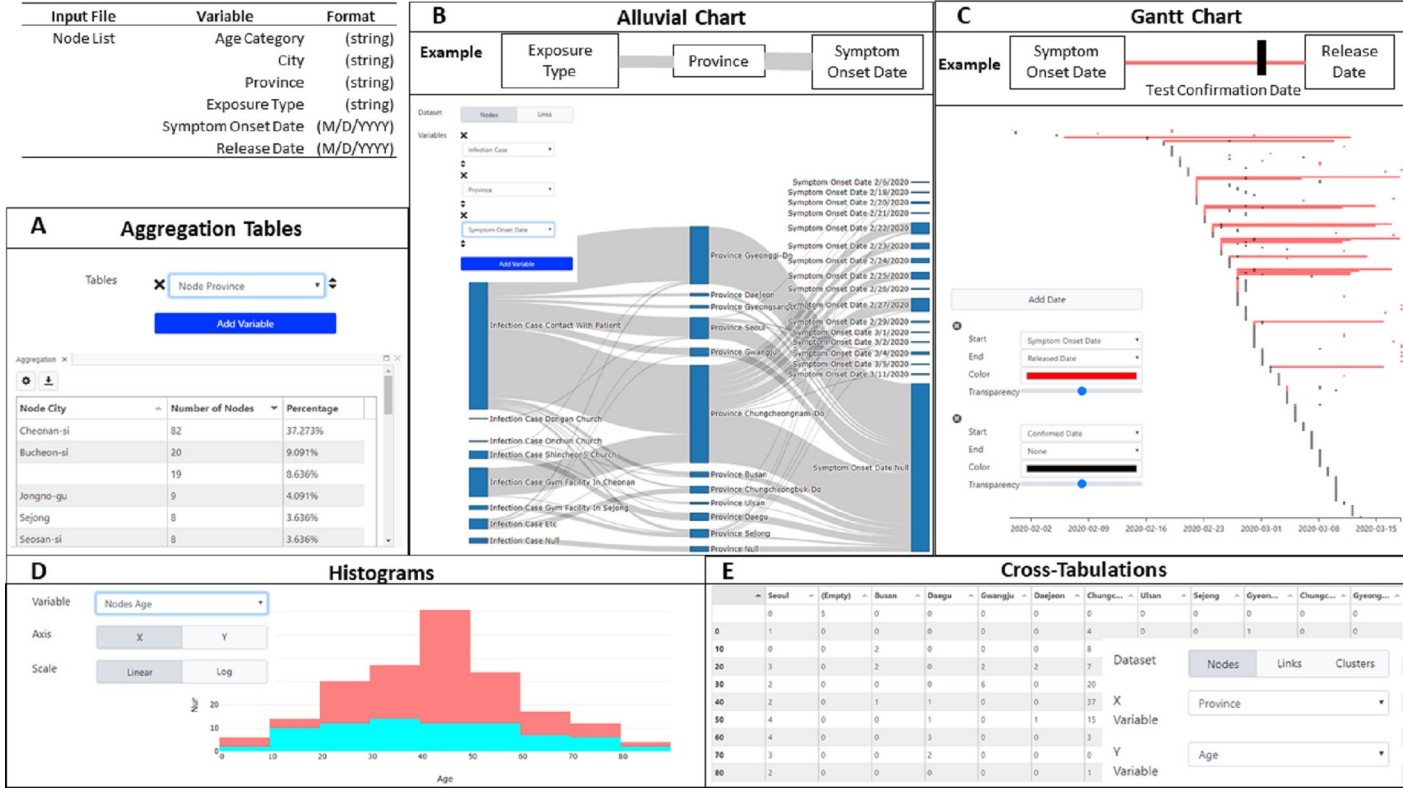

**Fig 4. MicrobeTrace visualization does not require genomic or contact tracing data and can easily calculate aggregation and cross-tabulation tables in addition to visualizing histograms, alluvial/flow diagrams and Gantt charts.** Each diagram has an inset settings menu that describes the settings changes necessary to achieve them. **(4A)** City-level aggregation achieved via a single dropdown selection. **(4B)** Alluvial diagram of associations between the Type of Exposure to COVID-19, Province, and Symptom Onset Date. **(4C)** Gantt charts to describe the span of time between Symptom Onset, Positive Test Confirmation, and Hospital Release Date. **(4D)** Age histogram, binned by decade and colored by gender. This histogram illustrates a trend identified early in the Korean COVID-19 outbreak, wherein a disproportionate number of middle-age female cases were diagnosed **(4E)** Cross-tabulation table of cases by City and Age categories.

facilitate exploration and communication of public health data. Each view is interactive and interoperable so nodes in one view are propagated to other views. For example, a node selected by search or click in the **Table View** is highlighted both there and in relevant adjacent views. Similarly, all choices on color-mappings for nodes and links are propagated to all other relevant views. All views are resizable and can be tiled to produce rich, interactive and exploratory dashboards as demonstrated below.

As with genetic data, networks are not required to leverage most of the visualizations in MicrobeTrace. Indeed, MicrobeTrace can be used to achieve rich visualizations using a list of nodes with a handful of variables like *age*, *gender*, *province*, *city*, *exposure type*, *symptom onset date*, *test confirmation date* and *hospital release data* available in the South Korea dataset. We demonstrate the construction of complex figures like a **Flow Diagram**, **Gantt Chart**, **Cross-tabulation**, **Aggregation**, and **Histogram** with simple dropdown menus (Fig 4). Additional diagrams can be achieved with the **2D Network**, **3D Network**, **Scatter Plot**, **Heatmap**, **Bubbles**, **Choropleth,** and **Globe Views** with relevant data types selected with simple dropdown menus. Operation of each view is documented in detail in the MicrobeTrace user manual (https://github.com/CDCgov/MicrobeTrace/raw/master/docs/MicrobeTrace_Manual.pdf)

The **Sequences View** can be used to export or check the quality of the pairwise alignment. The **Phylogenetic Tree View** will construct a tree via the neighbor-joining algorithm according to the provided pairwise distance calculations. The **Phylogenetic Tree View** has robust

customization controls that have been modularized in a separate JavaScript library called Tidy-Tree [36].

## Reproducibility

Public health investigations are iterative and the underlying data sources tend to grow over time. Once MicrobeTrace workspaces have been customized they can be saved in two ways: (1) as a custom MicrobeTrace file or (2) as a "stashed" (cached) browser session. As new data arrives, a user can choose to add new files and recompute the network while pinning nodes to their original positions on-screen. This capability enables a greater understanding of transmission dynamics by enforcing continuity between visualization and exploration sessions over time. Styling parameters and custom visualizations can be stored *independently* from the underlying data as a MicrobeTraceStyle file to facilitate communication between collaborators and preserve confidentiality. Style files can also be used to ensure continuity between public health investigations, such that different investigations yield identically styled visualizations even with different underlying data.

## Data and visualization exports

Communicating data from public health investigations is a complex process with messages being tuned to their audiences. To meet this need, MicrobeTrace is designed to provide users maximum control over visualization customization and export capabilities. For example, communication to academic and public health audiences often involves poster presentations that require images be scaled-up for large printer formats. We accommodate this requirement by enabling users to set specific export resolutions for PNG and JPEG formats or to export as Scalable Vector Graphics (SVGs) which can be easily enlarged without a loss of resolution. By default, a MicrobeTrace watermark is placed on images exported from MicrobeTrace; however, the transparency of the watermark can be increased using a menu slider to render it invisible. Taken together, these capabilities offer publication-ready image exports for scientific journals.

MicrobeTrace maximizes interoperability with other applications by enabling the export of all calculated and integrated datasets. The **Table View** renders tabular data which can be exported to comma-separated (CSV) and Excel (XLS, XLSX) file formats. The node-level table includes all information joined from multiple input data sources as well as calculated fields like a node's number of neighbors ('degree') and its cluster ID. The link-level table also includes calculated fields such as whether a link was identified as a 'nearest connected neighbor' as a Boolean result. MicrobeTrace offers robust filtering and selection capabilities that are also reflected in exported tables; 'Selected' and 'Visible' states are shown as Boolean results. Tables produced in the **Aggregation View** can be exported as formatted PDFs, CSVs, a zipped collection of CSVs, or an XLS/XLSX workbook where each aggregation is shown on independently named worksheets (Fig 4A). Data derived from the **Map**, **Globe**, and **Choropleth Views** can be exported as GeoJSON files for interoperability with other Geographic Information System (GIS) software. Genomic sequence alignments can be exported in FASTA or MEGA file formats in the **Sequences View.**

## Statistics and analysis of MicrobeTrace usage

We used Google Analytics for collection and analysis of MicrobeTrace website (https://microbetrace.cdc.gov) user traffic and activity. We categorized the data based on CDC users in Georgia, U.S. and worldwide non-CDC users. Most CDC users and MicrobeTrace developers are located in Atlanta suburbs and CDC campuses. MicrobeTrace trainings were excluded

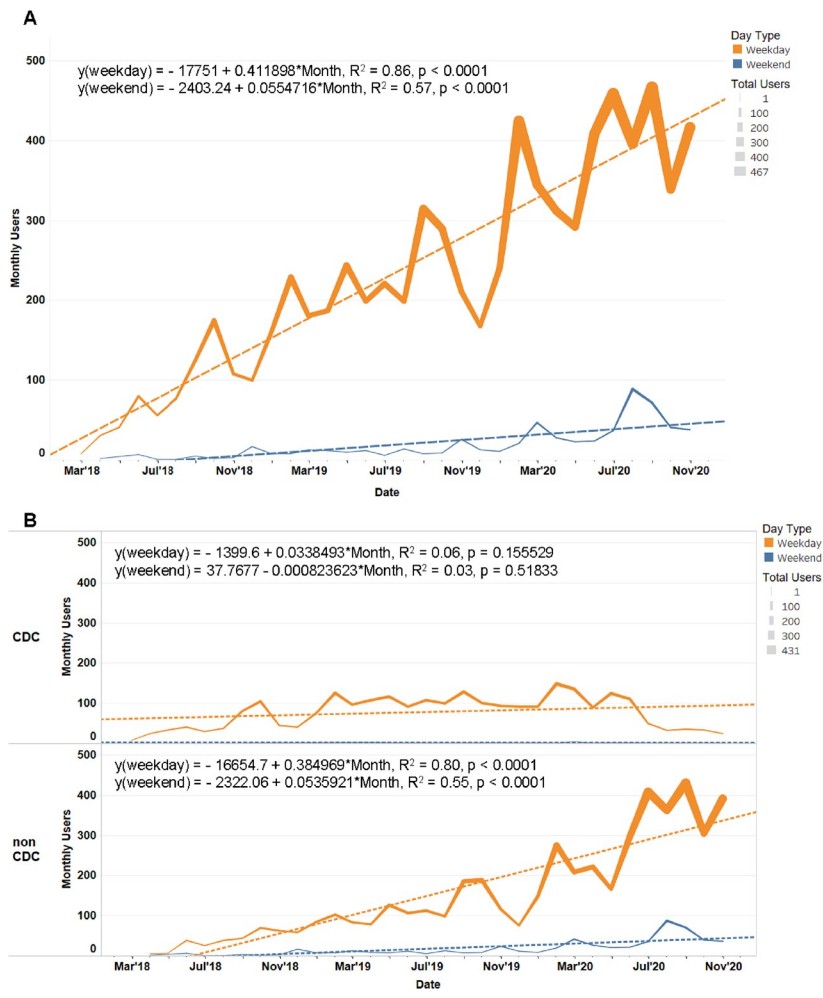

**Fig 5. MicrobeTrace's primary users are public health officials with the bulk of usage during the work week, as opposed to during the weekend.** In orange, are the number of monthly weekday users. In blue, are the number of monthly weekend users. Each month's mean daily user count is mapped and colored by day of the week. A lineal regression for each day type is shown to smooth the month-to-month effects and highlight the increasing usage trend. **(5A)** Monthly user traffic, stratified by weekday and weekend use. **(5B)** Monthly user traffic categorized by CDC and non-CDC traffic and stratified by weekday and weekend use.

from the data analysis (S1 Table). Statistical analyses and dashboards of MicrobeTrace monthly user trends (Fig 5) and geographic location (S1 Fig) were done with Tableau Software Desktop (version 2020.3.0). S1 Fig was created using MicrobeTrace usage data collected with Google Analytics which was mapped into the Tableau Software Light style built-in background map. Maps from Mapbox and OpenStreetMap are available by default in the Tableau Software Map Layers pane. Each Tableau Software map built-in includes acknowledgments of Mapbox (https://www.mapbox.com/tableau/) and OpenStreetMap (https://www.openstreetmap.org/). OpenStreetMap is free to use under an open license. Tableau Software was also used to compute linear trend models of weekday and weekend usage for the total number of users, and for CDC and non-CDC users. These models were estimated to be statistically significant at p< = 0.05.

Since the launch of MicrobeTrace in March 2018 until November 2020, a total of 4,119 (CDC, n = 921, 22%; non-CDC, n = 3198, 78%) unique users have accessed MicrobeTrace

about 11,194 times (mean user visits = 2.7). These visits amount to a combined 1,702 hours (mean visit duration = 8.7 minutes). MicrobeTrace usage is more frequent on weekdays than weekends, reflecting primary use by public health professionals. Monthly weekday users steadily increased since launch (0.41 new users/day) whereas weekend usage grew modestly over the same period (0.06 new users/day). User growth consisted largely of non-CDC users (0.38 new users/day, 93% of new user growth/weekday) while CDC user growth was relatively consistent (0.03 new users/day, 7% of new user growth/weekday).

Notably, at the beginning of the COVID-19 pandemic, a 78% user increase was observed from January (n = 251) to February 2020 (n = 446) when compared to the 40% user increase from January (n = 169) to February 2019 (n = 237). The average number of users per month also increased from 229 in 2019 to 411 in 2020, an increase of 80% (Fig 5A). In contrast to the summer of 2019, when a decrease in MicrobeTrace usage was seen, users increased 57% from May (n = 315) to July 2020 (n = 495). We also observed a notable increase of non-CDC users during the weekends of August (n = 89) and September 2020 (n = 72) (Fig 5B). The overall MicrobeTrace usage was consistently higher through November 2020.

## Summary

MicrobeTrace has been used to investigate a broad variety of infectious diseases, including CDC-assisted HIV cluster investigations in multiple states [37–40], investigations of hepatitis C virus (HCV) [41]), integrated into the CDC's Global Hepatitis Outbreak and Surveillance Technology (GHOST) used for viral hepatitis investigations [42] and is broadly used to integrate genomic and epidemiologic data for tuberculosis outbreak investigations [43]. It has also been used to integrate partner services, epidemiologic and whole genome data to better understand transmission during a retrospective public health investigation of *Neisseria gonorrhoeae* [30]. MicrobeTrace has also been used in foodborne pathogens outbreaks, such as *Escherichia coli O157*:H7 [44] and is currently being deployed for Ebola and SARS-CoV-2 outbreak investigations[45–48].

MicrobeTrace offers a suite of capabilities that are typically only achievable with an array of independent software, tools, and custom scripts, and substantive computational experience. A putative MicrobeTrace user typically achieves proficiency after one brief training session and aided by a cursory understanding of common browser interactions, such as 'drop-down menus', 'slider bars', and 'drag-and-drop'. Many standalone tools are available to calculate pairwise genetic distances with varying degrees of specificity to the pathogen of interest. MEGA (Molecular Evolutionary Genetics Analysis) is software broadly used in public health, but new users can be overwhelmed by dense interfaces with scores of options that are often dense with jargon and required inputs [49]. HIV-TRACE, which is specific to HIV sequence data, now offers rich visualization capabilities but its installation requires a keen understanding of Unix and the Git protocol for local installation and use [4]. Patristic distance calculations are available via the APE package in R or the Java application PATRISTIC, but these require programming expertise and software installations [25,50]. Integration and visualization of links with individual-level data can be a complex task requiring tools like Gephi or Cytoscape [6,9]. Even for those with programming expertise, options require use of decade-old libraries in R with the iGraph package or in Python with the NetworkX and MatPlotLib packages [51–53]. Even so, these visualizations are not interactive, do not provide additional figures, and must be augmented with separate network-level calculations and manipulations, all of which are easily performed in MicrobeTrace. Anecdotally, use of MicrobeTrace and its network layout interface can be playful; which has been shown to improve user experience and increase their motivation to use the tool [54].

While MicrobeTrace has been developed for public health, it also has many applications in academia, including arbitrary networks with independent node- and edge-level characteristics that are necessary to evaluate social, behavioral, biochemical, cellular, technological and physical networks. MicrobeTrace also offers rich customizations that reduce the time and effort to achieve insights with a novel data set. We are not aware of another tool that offers these capabilities in a secure, interoperable, and light-weight format that requires no installation prior to use.

## Availability and future directions

MicrobeTrace is an open source project that is anonymously available on a source repository hosted by GitHub at https://github.com/CDCgov/MicrobeTrace. The application is available at https://microbetrace.cdc.gov, accessible from a modern web browser and requires no installation to use. The 'readme' at the GitHub repository contains links for users such as a manual, example input files, training materials, and webinars. Example test data are located at https://github.com/CDCgov/MicrobeTrace/tree/master/demo. Multiple use cases are represented by these sample data, please reference Fig 1 for common data source pairings. The 'readme' at the GitHub repository also contains links and guides for prospective developers interested in contributing or developing a custom implementation.

MicrobeTrace is under active development for use during outbreak investigation and responses. The development team has a strong interest in improving interoperability with existing bioinformatic and public health surveillance platforms. The development team has also prioritized (1) optimizing sequence alignment methods, (2) inclusion of sequence quality control metrics, and (3) inclusion of algorithms commonly used in network analysis.

## Supporting information

**S1 Fig. Global map of unique MicrobeTrace users that have accessed the website at least once.** Color shading and marks are labeled by the total number of unique users from March 1st, 2018 to November 30th, 2020. The overwhelming majority of users access MicrobeTrace in the U.S. (84%) and international usage mostly comes from Vietnam (3%), China (2%), United Kingdom (1%), Australia (1%) and Canada (1%). Sixty-five additional countries have <1% users. Maps were created using Tableau.©Mapbox and ©OpenStreetMap are available by default in the Tableau Software® Map Layers pane. Each Tableau Software® map built-in includes acknowledgments of ©Mapbox (https://www.mapbox.com/tableau/) and ©OpenStreetMap (https://www.openstreetmap.org/). ©OpenStreetMap is free to use under an open license.
(TIFF)

**S2 Fig. Computational times for MicrobeTrace plotted against various input file types and numbers of taxa/sequences.**
(TIF)

**S1 Table. Number of MicrobeTrace users and activity collected from Google Analytics.**
(PDF)

**S2 Table. Scalability: time taken for MicrobeTrace to process different numbers of aligned HIV-1 polymerase (*pol)* sequences in a FASTA file to generate a network.**
(DOCX)

**S3 Table. Scalability: time taken for MicrobeTrace to process different numbers of aligned SARS-CoV-2 whole genome sequences in a FASTA file to generate a network.** Dashes in the

last row indicate that MicrobeTrace has reached the upper limit of processing, and is unable to compute a network.
(DOCX)

**S4 Table. Scalability: time for MicrobeTrace to process an imported Newick tree with different numbers of taxa to generate a network.**
(DOCX)

## Acknowledgments

We are thankful to our colleagues in the Division of Tuberculosis Elimination (Kathryn Winglee, Sarah Talarico, Yuri Springer, Benjamin Silk), the Division of STD Prevention (Kim Gernert, Katy Town, Matthew Schmerer), the Division of Viral Hepatitis (Seth Sims, Garrett Atkinson, Yury Khudyakov), National Center for HIV/AIDS, Viral Hepatitis, STD and TB Prevention—Informatics Office (Max Mirabito, Silver Wang), Transmission and Molecular Epidemiology Team (Alexandra Oster, Cheryl Ocfemia, Nivedha Panneer, Scott Cope, Sheryl Lyss) for providing valuable feedback, features, bug reports, and continued training of our public health partners. We are also thankful to our user base in public health and academia for reporting bugs and suggesting features with regularity.

## Disclaimers

Use of trade names is for identification only and does not imply endorsement by the U.S. Centers for Disease Control and Prevention (CDC). The findings and conclusions in this report are those of the authors and do not necessarily represent the views of the CDC.

## Author Contributions

**Conceptualization:** Ellsworth M. Campbell, Anthony Boyles, William M. Switzer.

**Investigation:** Ellsworth M. Campbell.

**Methodology:** Ellsworth M. Campbell, Anthony Boyles, Sergey Knyazev.

**Project administration:** William M. Switzer.

**Software:** Ellsworth M. Campbell, Anthony Boyles, Jay Kim, Sergey Knyazev.

**Supervision:** William M. Switzer.

**Validation:** Ellsworth M. Campbell, Anupama Shankar, Jay Kim.

**Writing – original draft:** Ellsworth M. Campbell, Anthony Boyles, Roxana Cintron, William M. Switzer.

**Writing – review & editing:** Ellsworth M. Campbell, Anupama Shankar, William M. Switzer.

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
