## [Decision Letter · Decision Letter 0]

2 May 2021

Dear Ms Shankar,

Thank you very much for submitting your manuscript "MicrobeTrace: Retooling Molecular Epidemiology for Rapid Public Health Response" for consideration at PLOS Computational Biology. As with all papers reviewed by the journal, your manuscript was reviewed by members of the editorial board and by several independent reviewers. The reviewers appreciated the attention to an important topic. Based on the reviews, we are likely to accept this manuscript for publication, providing that you modify the manuscript according to the review recommendations.

Sincerely,

Manja Marz

Software Editor

PLOS Computational Biology

Manja Marz

Software Editor

PLOS Computational Biology

[LINK]

Reviewer's Responses to Questions

**Comments to the Authors:**

Reviewer #1: The paper is well written and the description of the MicroTrace sufficiently detailed. I have no comments about the manuscript

Reviewer #2: Thank you for the opportunity to review “MicrobeTrace: Retooling Molecular Epidemiology for Rapid Public Health Response” by Campbell and colleagues. The authors describe MicrobeTrace, a browser-based graphical user interface for analyzing multiple data sources including epidemiological, genomic, and GIS data. They present the general capabilities and highlight the functionality via a number of applied vignettes. They also show statistics on usage since its implementation in 2018.

Overall, the manuscript is very polished and easy to read. The authors “hit the nail on the head”, so to speak, in terms of the difficulty for end users in public health to meaningfully use all of the various data that are currently being generated, most notably the avalanche of SARS-CoV-2 sequencing data that are flooding public repositories. MicrobeTrace appears very promising. I was able to access the site and easily upload and visualize my own data (nwk, fasta, and metadata files) without watching the tutorials. As such, I find it very intuitive (although I didn’t immediately see how to change the root of a phylogeny). Also, what I find was extremely interesting is the ability to create multi-layer networks using genomic and epidemiological data. I was not able to personally explore this functionality, but I trust it delivers as promised.

Nevertheless, there are a couple items that I found missing. 1) For one, the authors reference the other competing web applications in use, Nextstrain and MicroReact. One aspect that I appreciate on those platforms is the readily accessible vignettes that are pre-loaded in the database. This is especially prominent for MicroReact where a number of public projects are available. Researchers can even publish the project along with their publication for readers to navigate themselves. I was able to find the MicrobeTrace demo on the GitHub page, but I feel public vignettes would highlight the utility. 2) Is there the ability to access publicly available genomic data via MicrobeTrace, for example, by providing a list of accession numbers, or do these data have to be downloaded by the end user? These comments are more related to the functionality of MicrobeTrace and not necessarily the manuscript itself.

There are a few items that I do feel the authors should address in the manuscript.

1) What is the scalability of MicrobeTrace on a standard computer (e.g., Dell XPS or MacBook Pro)? I would have appreciated a figure that showed computation time with increasing numbers of sequencing or network data. For example, how many aligned SARS-CoV-2 genomes can the stand-alone version handle? How big (taxa/nodes) of a phylogeny or network can be rendered? What happens when you add GPS data? This performance data is crucial for analyzing the utility to the end user.

2) One of the most common questions that public health investigators ask the laboratorians (or the bioinformatician that is performing the genomic analysis) relates to inferred transmission events among cases. How closely related are the pathogen genomes? Does this mean that they are related in a transmission event? MicrobeTrace currently offers methods to set thresholds when viewing networks or phylogenies, but what can be done in terms of providing the end user a probability of transmission based on all of the epi and molecular/genomic data they are co-visualizing. Can algorithms be put in place to at least provide some guidance on the relatedness of cases? A number of researchers are working in this area (e.g., Xavier Didelot’s TransPhylo that uses epi and genomic data to assign probabilities for transmission events or Christoph Fraser’s PhyloScanner).

3) Throughout the development of MicrobeTrace, were end-users (county and state level epi staff on the investigations side) engaged to identify what data, reports, or visualizations are useful to them? If so, what was the feedback? If not, why and are there plans on doing so? I feel this is essential for ensuring widespread adoption and implementation. Perhaps canned reports could be made available based on the data types added?

4) Are the future plans to integrate MicrobeTrace with electronic laboratory and case reporting systems? This has been a huge bottleneck at the state health department level as many ELR and ECR systems are still poorly integrated. Now, as states add genomic data, there is no easy way to link it to case data. If would be great if MicrobeTrace could fill this need.

P.S. As an Epi Info user in the 2000’s, it’s amazing to see how far we have come!

**Have all data underlying the figures and results presented in the manuscript been provided?**

Reviewer #1: Yes

PLOS authors have the option to publish the peer review history of their article (what does this mean?). If published, this will include your full peer review and any attached files.

Reviewer #1: No

Reviewer #2: **Yes: **Taj Azarian

**Have the authors made all data and (if applicable) computational code underlying the findings in their manuscript fully available?**

Reviewer #2: Yes

Figure Files:

Data Requirements:

Reproducibility:

References:

---

## [Decision Letter · Decision Letter 1]

23 Jul 2021

Dear Ms Shankar,

We are pleased to inform you that your manuscript 'MicrobeTrace: Retooling Molecular Epidemiology for Rapid Public Health Response' has been provisionally accepted for publication in PLOS Computational Biology.

Best regards,

Manja Marz

Software Editor

PLOS Computational Biology

Manja Marz

Software Editor

PLOS Computational Biology

Reviewer's Responses to Questions

**Comments to the Authors:**

Reviewer #2: Thank you for the opportunity to review the revised manuscript. I appreciate the considerable effort of the authors in addressing my questions, especially since some were clearly outside the scope of the current manuscript and driven by my interest in their work. I do feel that the additional supplemental tables were very helpful in putting the computational time in scale - I had imagined they were orders of magnitude higher than what the authors presented in the revised version of the manuscript. Also, I feel the additional paragraph about the community contributions to the work is very valuable. That sort of engagement should be highlighted as exemplar in the field of tool development. I have no further comments at this time.

**Have the authors made all data and (if applicable) computational code underlying the findings in their manuscript fully available?**

Reviewer #2: Yes

PLOS authors have the option to publish the peer review history of their article (what does this mean?). If published, this will include your full peer review and any attached files.

Reviewer #2: **Yes: **Taj Azarian

---

## [Editor Report · Acceptance letter]

1 Sep 2021

PCOMPBIOL-D-21-00215R1 

MicrobeTrace: Retooling Molecular Epidemiology for Rapid Public Health Response

Dear Dr Shankar,

I am pleased to inform you that your manuscript has been formally accepted for publication in PLOS Computational Biology. Your manuscript is now with our production department and you will be notified of the publication date in due course.

With kind regards,

Katalin Szabo
